# Downregulation of STAT3 in Epstein-Barr Virus-Positive Hodgkin Lymphoma

**DOI:** 10.3390/biomedicines10071608

**Published:** 2022-07-06

**Authors:** Stefan Nagel, Corinna Meyer, Sonja Eberth, Josephine Haake, Claudia Pommerenke

**Affiliations:** Department of Human and Animal Cell Lines, Leibniz-Institute DSMZ, 38124 Braunschweig, Germany; cme@dsmz.de (C.M.); seb14@dsmz.de (S.E.); joh21@dsmz.de (J.H.); cpo14@dsmz.de (C.P.)

**Keywords:** ALCL, DLBCL, BAG1, CELF2, HIPK2, NFIB, ZFP64

## Abstract

STAT3 is a transcription factor which is activated via various signaling transduction pathways or Epstein-Barr virus (EBV) infection and plays an oncogenic role in lymphoid malignancies including Hodgkin lymphoma (HL). The tumor cells of HL are derived from germinal center B-cells and transformed by chromosomal rearrangements, aberrant signal transduction, deregulation of developmental transcription factors, and EBV activity. HL cell lines represent useful models to investigate molecular principles and deduced treatment options of this malignancy. Using cell line L-540, we have recently shown that constitutively activated STAT3 drives aberrant expression of hematopoietic NKL homeobox gene HLX. Here, we analyzed HL cell line AM-HLH which is EBV-positive but, nevertheless, HLX-negative. Consistently, AM-HLH expressed decreased levels of STAT3 proteins which were additionally inactivated and located in the cytoplasm. Combined genomic and expression profiling data revealed several amplified and overexpressed gene candidates involved in opposed regulation of STAT3 and EBV. Corresponding knockdown studies demonstrated that IRF4 and NFATC2 inhibited STAT3 expression. MIR155 (activated by STAT3) and SPIB (repressed by HLX) showed reduced and elevated expression levels in AM-HLH, respectively. However, treatment with IL6 or IL27 activated STAT3, elevated expression of HLX and MIR155, and inhibited IRF4. Taken together, this cell line deals with two conflicting oncogenic drivers, namely, JAK2-STAT3 signaling and EBV infection, but is sensitive to switch after cytokine stimulation. Thus, AM-HLH represents a unique cell line model to study the pathogenic roles of STAT3 and EBV and their therapeutic implications in HL.

## 1. Introduction

Hodgkin lymphoma (HL) is a germinal-center-derived B-cell malignancy, although rare cases with T-cell origin have been described [1]. The typical large tumor cells are called Hodgkin Reed Sternberg (HRS) cells in classical-HL- and lymphocyte-predominant (LP) cells in nodular-lymphocyte-predominant HL. In both subtypes, the tumor cells are rare in infiltrated lymph nodes, while most cells of the tumor mass represent reactive lymphocytes, macrophages, dendritic cells, and granulocytes [2]. The specific phenotype and rareness of HRS and LP cells complicate their analysis. However, established bona fide HL cell lines may serve as suitable models to analyze their molecular abnormalities and cellular characteristics [3]. Accordingly, several tumor hallmarks have been identified and analyzed in HL, including aberrant receptor signaling, activated NFkB pathway, inhibition of apoptosis, loss of B-cell phenotype, and immune escape [2,4]. Epstein-Barr virus (EBV) infection is frequently associated with HL and may play a role in early steps of lymphomagenesis [4]. EBV-encoded genes deregulate important factors involved in the pathogenesis of HL including PAX5, STAT3 and MIR155 [5,6,7,8,9]. Furthermore, in HL cells extensive chromosomal and copy number aberrations have been detected and particular genomic alterations mediate aberrant activation of oncogenes including REL and JAK2 [10,11,12,13].

HRS and LP cells lack typical B-cell characteristics including expression of B-cell-associated developmental transcription factors (TFs). Even the absence of master TFs PAX5 and EBF1 have been reported; however, their reactivation was insufficient to reconstitute the B-cell phenotype [14,15,16]. Thus, additional TFs are required to complete the disturbed B-cell differentiation. Analyzing the physiological activity of NKL homeobox genes in B-cell development revealed aberrant expression of HLX (H2.0-like homeobox) in HL, diffuse large B-cell lymphoma (DLBCL), and anaplastic large cell lymphoma (ALCL). HLX is a member of the NKL-code which comprises eleven hematopoietically expressed NKL homeobox genes [17]. HLX expression is normally restricted to early B-cell stages and deregulated by aberrantly activated STAT3 as described in HL, DLBCL and ALCL [18,19,20]. STAT3 is the executing component of the JAK-STAT-signaling pathway, which is driven by interleukines IL6 or IL27 [21,22]. IL6-signalling and JAK2 are aberrantly activated and overexpressed in HL [13,23]. Subsequently, activated and phosphorylated STAT3 translocate into the nucleus and regulate target genes including MIR155 and HLX [18,24,25]. Aberrantly activated or mutated STAT3 plays an oncogenic role in several lymphoid malignancies including HL, DLBCL and ALCL [26,27,28,29].

Here, we analyzed the EBV-positive HL cell line AM-HLH [30]. Interestingly, this cell line showed both reduced STAT3 activity and weak HLX expression. Our study revealed opposing roles of STAT3 in the regulation of oncogenes and EBV. Therefore, these data may have implications for therapeutic strategies in HL subsets.

## 2. Materials and Methods

### 2.1. Cell Lines and Treatments

Cell lines were obtained from the DSMZ (German Collection of Microorganisms and Cell Lines, Braunschweig, Germany): AM-HLH (ACC 908), HDLM-2 (ACC 17), KM-H2 (ACC 8), L-428 (ACC 197), L-540 (ACC 72), L-1236 (ACC 530), SUP-HD1 (ACC 574), U-HO1 (ACC 626). Cell culture conditions, culture media and other relevant information on each cell line are described elsewhere [30,31]. Analyzed cell lines have been validated for authentication and freedom from inadvertent mycoplasm and viral contamination.

Gene activities were suppressed via siRNA-mediated knockdown. Gene-specific siRNA oligonucleotides and AllStars negative control siRNA (siCTR) were purchased from Qiagen (Hilden, Germany). A green fluorescence protein (GFP)-labeled STAT3 expression construct was cloned into vector pCMV6-XL4 and obtained from Origene (Wiesbaden, Germany). SiRNAs (80 pmol) and plasmid DNA (2 µg) were transfected into 1 × 10^6^ cells by electroporation using the EPI-2500 impulse generator (Fischer, Heidelberg, Germany) at 350 V for 10 ms. Transfected cells were harvested after 20 h cultivation. Cell stimulations with 20 ng/mL interleukin (IL)6 or IL27 (R&D Systems, Wiesbaden, Germany) were performed for 4 h, and for 20 h with 10 µg/mL histone deacetylase inhibitor TSA (Sigma, Taufkirchen, Germany). All knockdown and stimulation experiments were performed twice generating similar results.

For functional testing, treated cells were analyzed with the IncuCyte S3 Live-Cell Analysis System (Essen Bioscience, Hertfordshire, UK). Apoptotic cells were quantified using the IncuCyte Caspase-3/7 Green Apoptosis Assay diluted at 1:2000 (Essen Bioscience, Hertfordshire, UK). Live-cell imaging experiments were performed twice with four-fold parallel tests.

### 2.2. Polymerase Chain-Reaction (PCR) Analyses

Total RNA was extracted from cell line samples using TRIzol reagent (Invitrogen, Darmstadt, Germany). cDNA was synthesized by random priming from 5 µg RNA using Superscript II (Invitrogene, Darmstadt, Germany). Expression analysis of EBV-encoded genes LMP1, LMP2A, EBNA1, EBNA2, EBNA3A, EBNA3C and of control gene YY1 was performed as described previously [19]. Reverse transcription (RT)-PCR was performed using taqpol (Qiagen) and the thermocycler TGradient (Biometra, Göttingen, Germany). Oligonucleotides were obtained from Eurofins MWG (Ebersberg, Germany).

Real-time quantitative (RQ)-PCR analysis was performed with the 7500 Real-time System using commercial buffer and primer sets (Thermo Fisher Scientific, Darmstadt, Germany). For normalization of expression levels, we analyzed the transcript of TATA box binding protein (TBP). We used the ddCT-method, and the obtained values are indicated as fold expression in relation to one selected sample which was set to 1. Quantitative PCR analyses were performed in triplicate. Standard deviations are calculated for each experiment and presented in the figures as error bars. Statistical significance was assessed by *t*-test and the derived *p*-values are indicated by asterisks (* *p* < 0.05, ** *p* < 0.01, *** *p* < 0.001, n.s. not significant).

### 2.3. Protein Analyses

Western blots were generated using the semi-dry method. Protein lysates from cell lines were prepared using SIGMAFast protease inhibitor cocktail (Sigma). Proteins were transferred onto nitrocellulose membranes (Bio-Rad, München, Germany) and blocked with 5% dry milk powder dissolved in phosphate-buffered saline buffer (PBS). The following antibodies were used: alpha-Tubulin (Sigma), STAT3 (Cell Signalling, Leiden, Netherlands), phospho-(P)-STAT3 (Cell Signalling), and IRF4 (Origene). For loading controls, blots were reversibly stained with Poinceau (Sigma), and the detection of alpha-Tubulin (TUBA) was performed thereafter. Secondary antibodies (Southern Biotech, Birmingham, AL, USA) were linked to peroxidase for detection by Western-Lightning-ECL (Perkin Elmer, Waltham, MA, USA). Documentation was recorded using the digital system ChemoStar Imager (INTAS, Göttingen, Germany).

Immuno-cytology was performed as follows: cells were spun onto slides, air-dried and fixed with methanol/acetic acid for 90 s. The STAT3 antibody (see above) was diluted 1:20 in PBS containing 5% BSA and incubated for 30 min. Washing was performed 3 times with PBS. Preparations were incubated with fluorescent secondary antibody (diluted 1:100) for 20 min. After final washing the cells were mounted in Vectashield (Vector Laboratories, Burlingame, CA, USA), containing DAPI for nuclear staining. Images were captured with an Axion A1 microscope using Axiocam 208 color and software ZEN 3.3 blue edition (Zeiss, Göttingen, Germany).

### 2.4. Genomic Profiling Analysis

For genomic profiling by cytogenetic microarray, the genomic DNA of cell line AM-HLH was prepared by the Qiagen Gentra Puregene Kit (Qiagen). The labelling, hybridization and scanning of HD Cytoscan arrays was performed at the Genome Analytics Facility, Helmholtz Centre for Infection Research (Braunschweig, Germany), using HD arrays according to the manufacturer’s protocols (Affymetrix, High Wycombe, UK). Data were visualized and interpreted using the Chromosome Analysis Suite, software version 2.0.1.2 (Affymetrix). 

### 2.5. Gene Expression Profiling Analysis and Transcriptome Data

Gene expression profiling data from cell lines were generated at the Genome Analytics Facility using HG U133 Plus 2.0 gene chips (Affymetrix, High Wycombe, UK), as reported previously [32]. Expression profiling datasets for HL cell lines (GSE204717) and patients (GSE12453) were, respectively, deposited and obtained from GEO (https://www.ncbi.nlm.nih.gov/gds/, accessed on 1 February 2022) [33]. Data processing was performed via R/Bioconductor using limma and affy packages. Graphical presentation of the expression profiling data by heatmaps was performed using Heatmapper [34]. Transcriptome data for normal hematopoietic cells were obtained from The Human Protein Atlas [35].

## 3. Results

### 3.1. HL Cell Line AM-HLH Is EBV-Positive and HLX-Negative

In HL, DLBCL and ALCL, STAT3 aberrantly activates NKL homeobox gene HLX which plays an oncogenic role in these lymphoid malignancies [18,19,20]. Accordingly, we have shown that EBV-mediated activation of STAT3 drives the expression of HLX in DLBCL [19]. AM-HLH represents a well characterized and validated EBV-positive HL cell line [30]. Therefore, we used this model system to investigate expression and regulation of STAT3 and HLX in EBV-positive HL cells. RT-PCR analysis confirmed expression of several EBV-encoded genes in AM-HLH, including LMP1, LMP2A, EBNA1, EBNA2, EBNA3A and EBNA3C (Figure 1A). Consistently, the cells grow slightly clumpy (Figure 1B), as generally reported for cultures of EBV-infected B-cell lines [19,36,37]. Thus, our observations of AM-HLH confirmed the major indications of EBV infection.

However, RQ-PCR analysis of NKL homeobox gene HLX in cell lines derived from HL, DLBCL and ALCL detected only weak expression levels in AM-HLH. This result contrasted to EBV-negative HL cell line L-540, EBV-positive DLBCL cell line DOHH-2 and all examined EBV-negative ALCL cell lines (Figure 1C). HLX is a direct target gene of STAT3, which was shown to be constitutively active in L-540, activated by EBV in DOHH-2, and amplified and activated by fusion protein NPM1-ALK in ALCL cell lines [18,19,20,26]. Therefore, we then focused our interest on the regulation and activity of STAT3 in AM-HLH. 

### 3.2. Analysis of STAT3 in HL Cell Lines AM-HLH and L-540

RQ-PCR analysis of STAT3 in eight HL cell lines showed low expression levels in AM-HLH while HDLM-2, L-428 and L-540 expressed the highest levels (Figure 2A). Accordingly, Western blot analysis showed prominent STAT3 signals in L-428 and L-540 while AM-HLH expressed low protein levels (Figure 2A). Phosphorylated STAT3 was only detected in L-540 (Figure 2A) which reportedly expressed constitutively activated STAT3 [18,26]. In correspondence to its inactivated form, immuno-cytological analysis of STAT3 in AM-HLH demonstrated that this TF was mainly located in the cytoplasm (Figure 2B). In addition, cytological analysis of AM-HLH after forced expression of GFP-labeled STAT3 also showed absence of GFP-STAT3 in the nucleus (Figure 2B). Thus, AM-HLH expressed reduced STAT3 levels of RNA and protein, the latter of which was not activated by phosphorylation and located mainly in the cytoplasm. 

However, the treatment of AM-HLH with IL6 for 4 h resulted in STAT3 phosphorylation and translocation from the cytoplasm into the nucleus (Figure 2C), demonstrating its potential for activation in these cells. Furthermore, RQ-PCR analysis of HLX showed increased expression levels, confirming functional STAT3 activation (Figure 2C).

The treatment of L-540 cells with the deacetylase inhibitor TSA resulted in the translocation of STAT3 from the nucleus into the cytoplasm (Figure 2D). Accordingly, this treatment resulted in a reduced expression of HLX (Figure 2D), as described recently [18]. In contrast, the treatment of AM-HLH with TSA resulted in an enhanced HLX expression (Figure 2D). These data indicated that these cell lines differ in HLX regulation via protein-acetylation. In L-540, acetylated STAT3 is translocated into the cytoplasm and, hence, unable to activate HLX [18], while in AM-HLH, histone acetylation may, rather, play a more prominent role in HLX activation, possibly operating via chromatin remodeling. Taken together, TF STAT3 activated HLX in both L-540 and AM-HLH which, however, differ in STAT3 expression and regulation.

### 3.3. Analysis of Genomic Amplifications in AM-HLH

Genomic aberrations play a basic role in the pathogenesis of HL. Accordingly, the tumor cells contain several chromosomal rearrangements and copy number alterations as described in patients and cell lines [10,11,12,13]. The karyotype of AM-HLH has been established previously, showing many, albeit no, recurrent rearrangements [30]. Here, we extended these data of AM-HLH performing genomic profiling by cytogenetic microarray to map and quantify copy number alterations. The results for all chromosomes are shown in Appendix A. Interestingly, AM-HLH contained several amplifications showing extraordinarily high copy numbers. Therefore, we speculated that these aberrations may underlie our conflicting findings of EBV infection and reduced STAT3 activity.

We focused on seven amplicons exhibiting at least a 10-fold rise in their copy numbers which were located at 6p25, 9p24, 9p13, 16p13, 16p12, 16p11 and 20q13 (Appendix A). Gene expression profiling data were used to analyze expression levels of the amplified genes. The corresponding gene expression levels from AM-HLH and eight HL control cell lines were compared in tables and visualized in heatmaps (Appendix A and Figure 3). Genes expressed more than 4-fold in AM-HLH were indicated and promising candidates selected.

IRF4 encodes a hematopoietic TF and was amplified at 6p25 and overexpressed in AM-HLH as confirmed by RQ-PCR and Western blot analysis (Figure 3A). IRF4 has been shown to regulate STAT3 expression in ALCL which may operate in HL as well [38]. JAK2 encodes a receptor-associated kinase which mediates phosphorylation and activation of STAT3 [21]. Furthermore, JAK2 is a reported oncogene and overexpressed in HL by gene amplification [13]. Interestingly, the locus of JAK2 was located in the amplicon at 9p24 in AM-HLH but showed reduced expression levels. RQ-PCR analysis confirmed low expression in AM-HLH while HDLM-2 and L-1236 expressed the highest levels (Figure 3B). This finding may support the conclusion that HL cell line AM-HLH somehow escapes STAT3-activation. 

BAG1 encodes an activator of survival factor BCL2 [39]. BAG1 was amplified at 9p13 and overexpressed in AM-HLH. RQ-PCR analysis confirmed elevated expression levels in AM-HLH but also showed high levels in DEV and SUP-HD1 (Figure 3B). PAX5 encodes a TF operating as master B-cell factor [40]. PAX5 is frequently downregulated in HL, thus inhibiting B-cell differentiation in the tumor cells [14]. The gene of PAX5 is located in the amplicon at 9p13. However, RQ-PCR analysis of PAX5 showed for AM-HLH neither elevated expression (as observed in DEV) nor suppression (KM-H2, L-428 and L-540) (Figure 3B). PAX5 is activated by STAT3 and plays a regulatory role in EBV infection of B-cells [5,41].

The amplifications at 16p harbor several overexpressed genes which, however, indicates no connection to STAT3 or EBV (Figure 3C). In contrast, NFATC2 and ZFP64 encode hematopoietic TFs and were amplified at 20q13. NFATC2 is described as STAT3 regulator while ZFP64 activates KMT2A/MLL, representing a chromatin and general gene activator [42,43]. RQ-PCR analyses confirmed their exclusive overexpression in AM-HLH (Figure 3D).

To validate these findings and to identify additional genes differentially expressed in AM-HLH, we performed genome-wide comparative gene expression profiling analysis, analyzing the top-1000 up/downregulated genes (Appendix A). Accordingly, in comparison to eight control HL cell lines, AM-HLH overexpressed IRF4, BAG1, NFATC2 and ZFP64, in addition to KMT2A, CELF2, IL6R and IL27RA. Furthermore, AM-HLH showed reduced expression of HIPK2, NFIB, MIR155 and IL6. Thus, STAT3-signaling via IL6 and IL27 is possible but not conducted. HIPK2 and NFIB are described as STAT3 activators but were downregulated in AM-HLH [44,45]. MIR155 is a reported target gene of STAT3 and was also downregulated in AM-HLH [24,25]. Furthermore, MIR155 is activated by EBV and inhibited by CELF2 [46,47]. In the following, we used knockdown and stimulation experiments to analyze regulatory relationships of selected candidates in STAT3 regulation.

### 3.4. Impacts of Identified Gene Candidates in STAT3 Regulation

Our approach analyzing genomic and gene expression profiling data from AM-HLH and HL control cell lines revealed interesting candidates potentially involved in STAT3 regulation or lymphomagenesis. These genes and their downstream effects were analyzed by the modulation of their activity. SiRNA-mediated knockdown of IRF4 was confirmed by RQ-PCR and Western blot analysis (Figure 4A). The data indicated that IRF4 inhibited the expression of STAT3 and NFIB and activated BAG1 (Figure 4A). NFATC inhibited STAT3 as well, albeit less strong as IRF4, and activated MIR155-inhibitor CELF2 (Figure 4B).

Both, IL6- and IL27-signalling activates STAT3 [21,22]. Accordingly, treatment of AM-HLH with IL27 for 4 h resulted in phosphorylation of STAT3 as already shown for IL6 (Figure 4C). Concomitantly, the expression levels of MIR155 and HLX went up while IRF4 decreased. Treatment with IL6 resulted in similarly increased expression of MIR155 and decreased IRF4 (Figure 4C). Corresponding knockdown studies showed that IL27RA activated the STAT3 target genes MIR155 and HLX (Figure 4D). HLX in turn inhibited SPIB and BCL11A in AM-HLH as analyzed after HLX knockdown (Figure 4E) and described in L-540 [18].

KMT2A is a direct target of amplified Zinc-finger TF ZFP64 [43]. KMT2A in turn activated in AM-HLH the expression of CELF2 while leaving IRF4 and HIPK2 unaltered (Figure 4F). HIPK2 expression was downregulated in AM-HLH but elevated in HLX-positive L-540 (Appendix A). Knockdown of HIPK2 in L-540 resulted in HLX suppression (Figure 4G), supporting its reported role in STAT3 regulation [44]. Finally, knockdown of STAT3 showed that this TF activated in AM-HLH the genes MIR155 and PAX5 while IRF4 was slightly inhibited (Figure 4G).

Taken together, these data showed that overexpressed IRF4 and NFATC act as STAT3 inhibitors. Nevertheless, STAT3 was still ready for reactivation as demonstrated by treatment with IL6 or IL27. Furthermore, we detected downregulation of STAT3-activators NFIB and HIPK2 and of STAT3-target genes HLX, MIR155 and PAX5. CELF2 is a reported inhibitor of MIR155 and was activated by overexpressed NFATC and KMT2A.

### 3.5. Functional Analysis of IRF4 in AM-HLH

We identified IRF4 as an amplified and overexpressed STAT3-inhibitor in AM-HLH. To examine its potential oncogenic activity in AM-HLH we performed live-cell-imaging after siRNA-mediated knockdown of IRF4. The results indicated that IRF4 supported proliferation (Figure 5A) and inhibited apoptosis (Figure 5B). Thus, although IRF4 inhibited the expression of oncogene STAT3, this TF operated oncogenically. Accordingly, our finding that IRF4 activated BCL2-activator BAG1 (Figure 4A) may underlie its anti-apoptotic activity in AM-HLH. Thus, IRF4 may represent a central factor in the conflicting behavior of EBV-positive AM-HLH cells, showing reduced STAT3 activity. 

## 4. Discussion

The results of this study are summarized in Figure 6, illustrating regulation and downstream activities of STAT3 in the context of EBV-positive HL cells. Using AM-HLH as EBV-positive HL cell line model we showed that STAT3 is only weakly expressed, the protein is inactive, and located mainly in the cytoplasm. Consistently, STAT3 target genes HLX and MIR155 were downregulated. Analysis of expression and genomic profiling data revealed both, upstream regulators and genes located downstream of STAT3.

IRF4 played a central role in STAT3 inhibition. Its gene was amplified at 6p25 and consequently overexpressed. IRF4 also inhibited the expression of NFIB which has been shown to activate STAT3 [45]. Functional analysis demonstrated that IRF4 drives proliferation and survival (probably via BCL2-activator BAG1). Thus, IRF4 performed as a strong oncogene in AM-HLH. HIPK2 represented an additional downregulated STAT3 activator in AM-HLH [44]. Furthermore, NFATC2 was also operated as a STAT3 inhibitor and was overexpressed by amplification at 20q13.

JAK2 represents a major STAT3-activator and was downregulated in AM-HLH although located in an amplicon. This observation indicated that this cell line escapes STAT3-activation. As compared to HLX-positive cell line L-540, HLX target gene SPIB was elevated in AM-HLH. SPIB plays an important role for both, B-cell differentiation and EBV activity, thus, indicating a conflict of two pathognomonic mechanisms [48,49,50]. PAX5 is indirectly regulated by STAT3 and plays a dual role in B-cell differentiation and EBV function as well [5,51,52]. In AM-HLH, PAX5 was suppressed by STAT3-signalling and amplified at 9p13. Collectively, these data indicate opposing mechanisms operating in AM-HLH: oncogenic STAT3-pathway and oncogenic EBV infection. Although EBV reportedly activates STAT3, inhibition of STAT3 may support EBV activity in AM-HLH. 

Additional deregulated players identified in AM-HLH included ZFP64, KMT2A, CELF2, and MIR155. ZFP64 was amplified at 20q13 and overexpressed. This TF has been described as potent activator of KMT2A transcription [43], which was accordingly overexpressed in AM-HLH as well. Our analyses showed that both, NFATC2 and KMT2A, mediated activation of CELF2. CELF2, in turn, represents an inhibitor of MIR155 [47], which showed low expression levels in AM-HLH, possibly due to downregulation and inactivation of STAT3. Thus, inhibition of MIR155 occurred twice, via IRF4-STAT3 and CELF2. However, MIR155 is reportedly highly expressed in HL and activated by EBV [46,53]. Nevertheless, MIR155 suppresses the oncogenic immune escape via inhibition of PDL1 [54,55]. Thus, MIR155 performs conflicting functions in lymphomagenesis which may underlie its downregulation in HL cell line AM-HLH.

## 5. Conclusions

Using cell line AM-HLH as a model for EBV-positive HL, we recognized downregulated STAT3 activity. Amplified and overexpressed IRF4 played a major role in STAT3 inhibition. Genes located downstream of STAT3 have been described as both oncogenes and regulators of EBV. STAT3 was still ready for reactivation via IL6- and IL27-signalling, indicating a possible switch in EBV activity. Thus, our study revealed conflicting roles of STAT3 in HL and showed that AM-HLH represents a unique cell line model to gauge therapeutic options for EBV-positive HL patients.

## Figures and Tables

**Figure 1 biomedicines-10-01608-f001:**
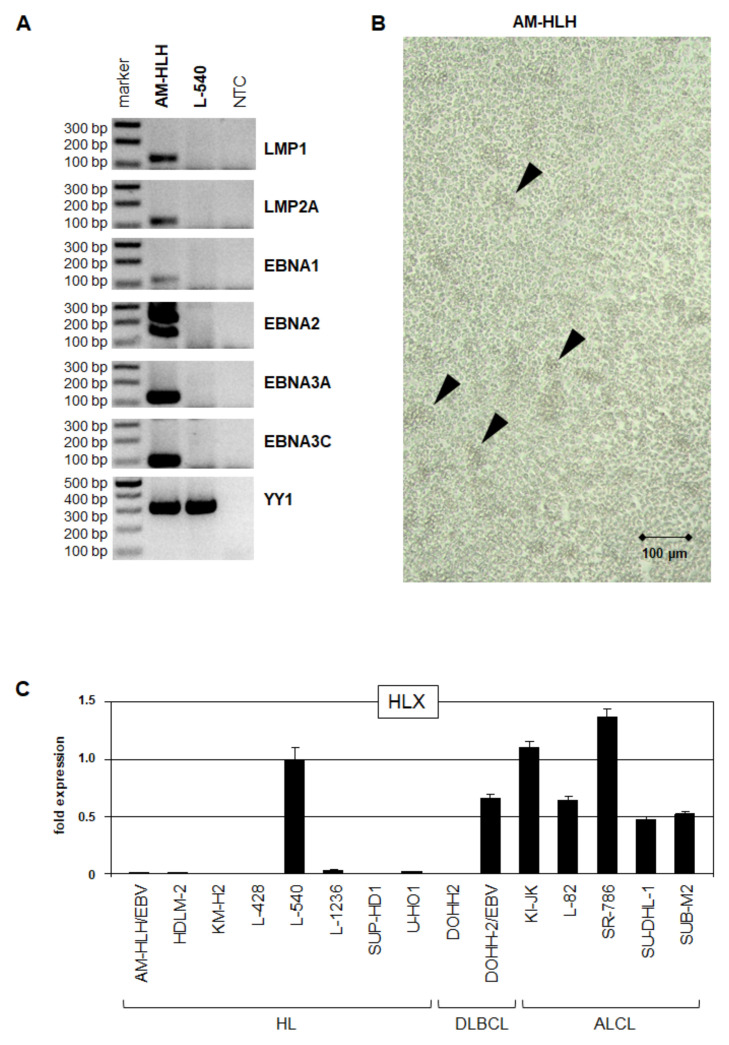
Expression of EBV and HLX transcripts in HL cell lines. (**A**) RT-PCR analysis of selected EBV genes in HL cell lines AM-HLH and L-540. YY1 was used as control. NTC: no template control. (**B**) Phase-contrast microscopical picture of AM-HLH cells in culture. Please note the appearance of cell aggregates which are indicative for EBV infection (arrow heads). (**C**) RQ-PCR analysis of NKL homeobox gene HLX in cell lines derived from HL, DLBCL and ALCL. Of note, we analyzed two reported DOHH-2 subclones—one was EBV-negative, the other EBV-positive.

**Figure 2 biomedicines-10-01608-f002:**
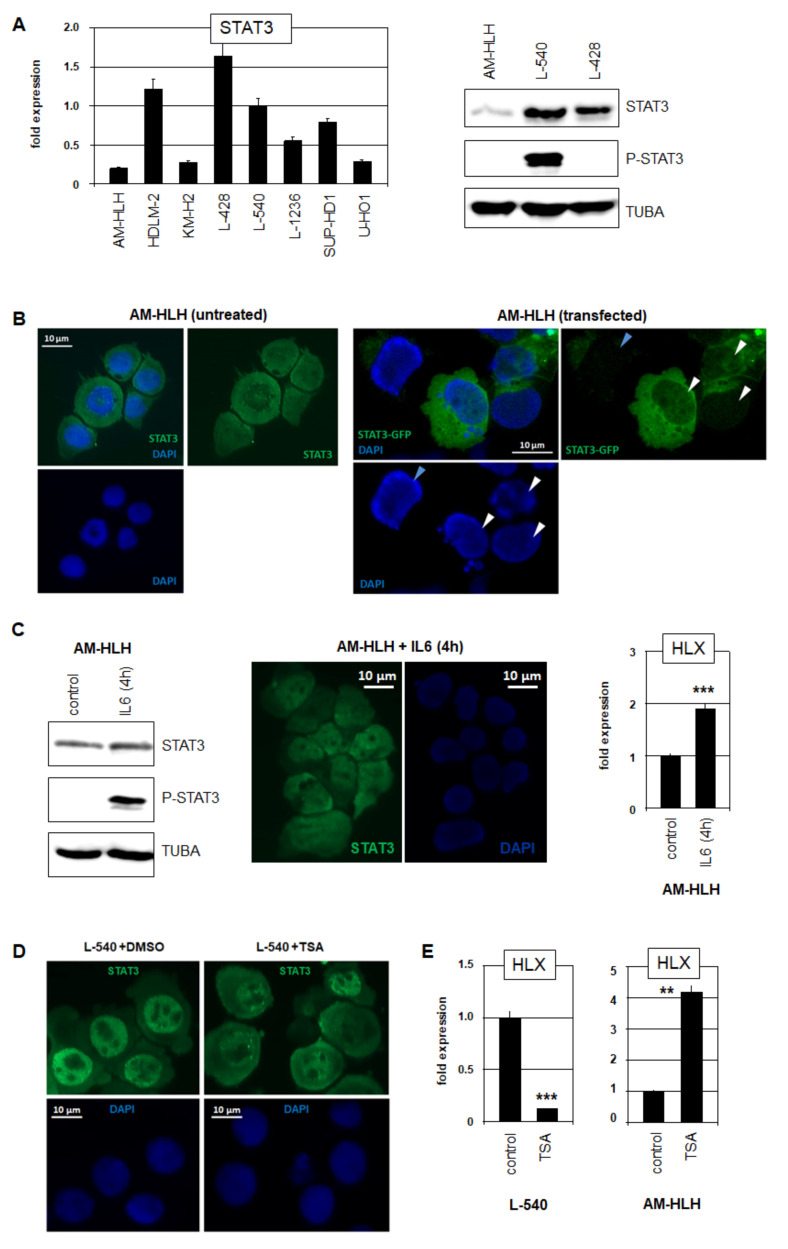
Analysis of STAT3 in HL cell lines. (**A**) RQ-PCR analysis of STAT3 in eight HL cell lines (left). Western blot analysis of three HL cell lines for STAT3, phospho (P)-STAT3 and alpha-tubulin (TUBA) as control (right). (**B**) Immuno-cytological microscopy of untreated AM-HLH cells (left) stained for STAT3 (green) and the nucleus (blue). Forced expression of GFP-labeled STAT3 in transfected AM-HLH (right). White arrowheads indicate transfected nuclei demonstrating reduced STAT3 protein in the nucleus as compared to the cytoplasm. The light blue arrowhead indicates a non-transfected cell. (**C**) Western blot analysis of IL6-treated AM-HLH for STAT3, P-STAT3 and TUBA (left). Immuno-cytological microscopy of IL6-treated AM-HLH cells stained for STAT3 (green) and the nucleus (blue) (middle). RQ-PCR analysis of HLX in IL6-treated AM-HLH after 4 h (right). (**D**) Immuno-cytological microscopy of control and TSA-treated L-540 cells stained for STAT3 (green) and the nucleus (blue). (**E**) RQ-PCR analysis of HLX in L-540 (left) AM-HLH (right) treated with TSA. Statistical significance was assessed by *t*-test and derived *p*-values indicated by asterisks (** *p* < 0.01, *** *p* < 0.001, n.s., not significant).

**Figure 3 biomedicines-10-01608-f003:**
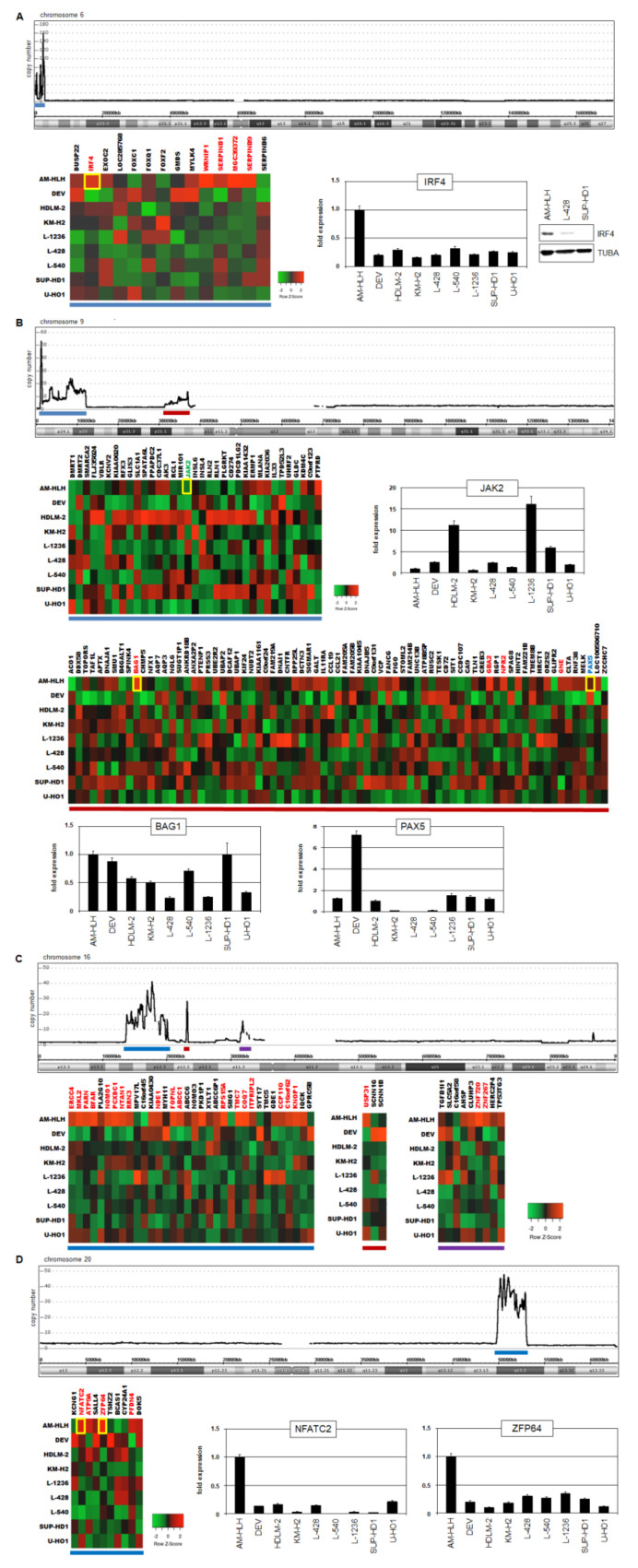
Genomic amplifications in AM-HLH and their consequences for gene activities. (**A**) Genomic profiling revealed copy number alterations for chromosome 6, indicating extraordinary amplifications at 6p25 (above). A heatmap shows gene expression values according to gene expression profiling data for 9 HL cell lines including AM-HLH. The blue line indicates the extent and location of analyzed genes. Expression of IRF4 in HL cell lines was analyzed by RQ-PCR and Western blot (below, right). (**B**) Copy number data for chromosome 9 indicate amplifications at 9p24 and 9p13 which are marked by blue and red lines, respectively (above). Heat maps showing the respective gene expression levels are marked with the corresponding colored lines and are shown below. RQ-PCR analyses in HL cell lines were performed for JAK2, BAG1 and PAX5. (**C**) Copy number data for chromosome 16 indicate amplifications at 16p13, 16p12 and 16p11 which are marked by blue, red and purple lines, respectively (above). Heat maps showing the respective gene expression levels are marked with the corresponding lines and are shown below. (**D**) Copy number data for chromosome 20 indicate an amplification at 20q13 which is marked by a blue line (above). A heat map shows the respective gene expression levels and is marked with the corresponding line (below). RQ-PCR analyses in HL cell lines were performed for NFATC and ZFP64.

**Figure 4 biomedicines-10-01608-f004:**
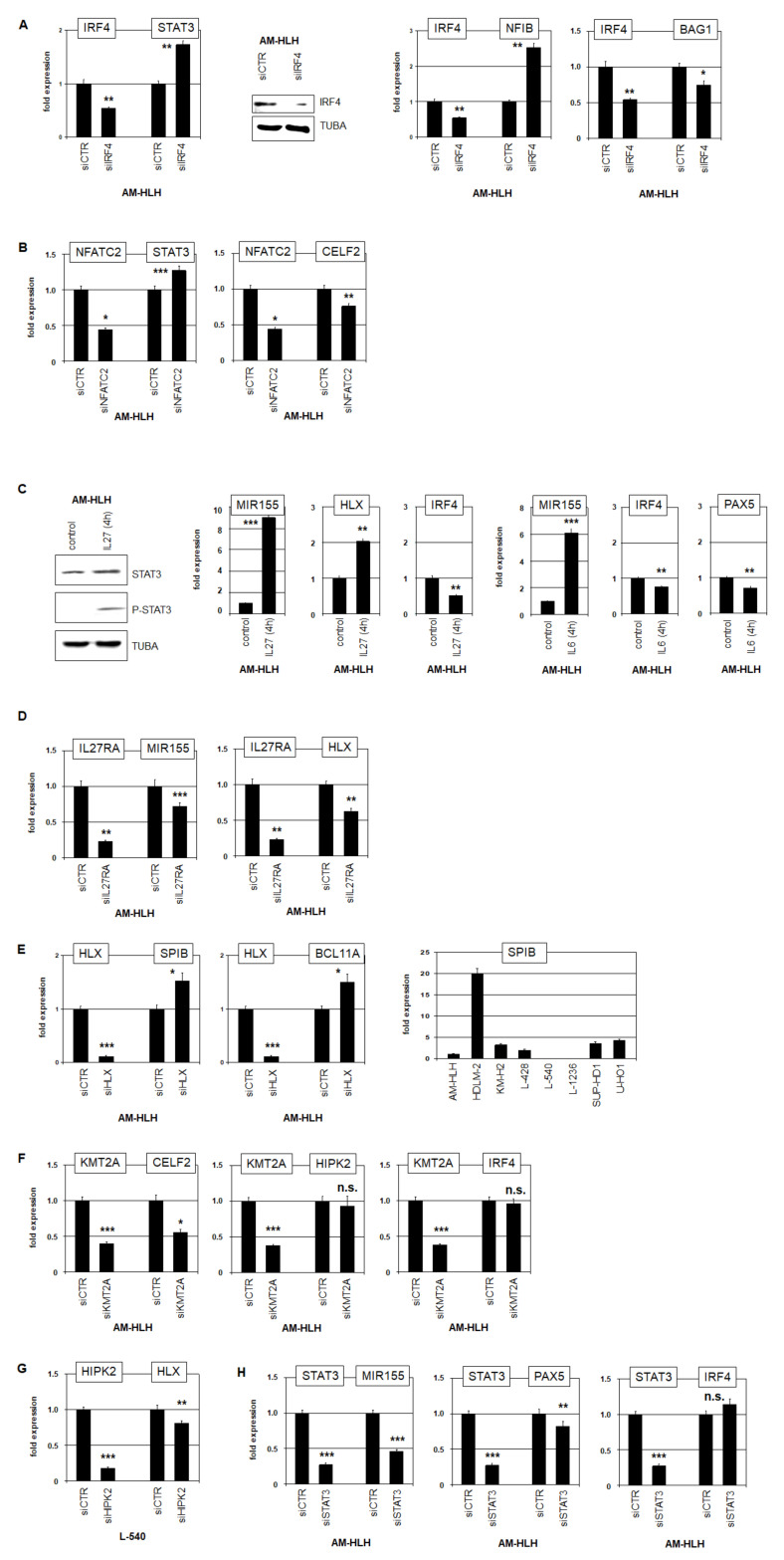
Knockdown and stimulation studies in AM-HLH and L-540. (**A**) SiRNA-mediated knockdown of IRF4 in AM-HLH resulted in elevated expression of STAT3 as analyzed by RQ-PCR (left). The knockdown of IRF4 was confirmed at the protein level by Western blot using TUBA as control (middle). Knockdown of IRF4 resulted in elevated expression of NFIB and reduced expression of BAG1 (right). (**B**) Knockdown of NFATC in AM-HLH resulted in slightly elevated expression of STAT3 and in reduced expression of CELF2. (**C**) Stimulation of AM-HLH with IL27 for 4 h mediated phosphorylation of STAT3 as shown by Western blot using STAT3 and TUBA as controls (left). This stimulation activated expression of MIR155 and HLX and inhibited expression of IRF4, as analyzed by RQ-PCR (middle). Stimulation with IL6 activated MIR155 and inhibited IRF4 and PAX5 (right). (**D**) Knockdown of IL27RA in AM-HLH resulted in downregulation of MIR155 and HLX. (**E**) Knockdown of HLX in AM-HLH resulted in elevated expression of SPIB and BCL11A (left). RQ-PCR analysis of SPIB expression in HL cell lines (right). (**F**) Knock down of KMT2A/MLL in AM-HLH resulted in reduced expression of CELF2, while the expression of HIPK2 and IRF4 was not altered. (**G**) Knock down of HIPK2 in L-540 resulted in reduced expression of HLX. (**H**) Knockdown of STAT3 in AM-HLH resulted in reduced expression of MIR155 and PAX5 and insignificantly elevated expression of IRF4. Statistical significance was assessed by *t*-test and derived *p*-values indicated by asterisks (* *p* < 0.05, ** *p* < 0.01, *** *p* < 0.001, n.s., not significant).

**Figure 5 biomedicines-10-01608-f005:**
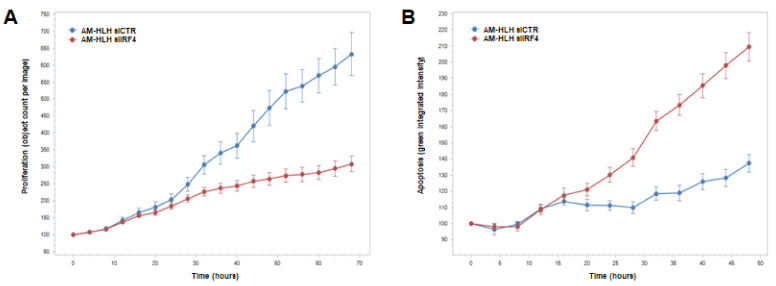
IRF4 impacts proliferation and apoptosis in AM-HLH. (**A**) SiRNA-mediated knockdown of IRF4 resulted in reduced proliferation while (**B**) apoptosis was enhanced. The data were generated by live-cell-imaging. Standard deviations are indicated in the figure, the *p*-values for the last points of time are *p* = 0.000338 and *p* = 0.00208, respectively.

**Figure 6 biomedicines-10-01608-f006:**
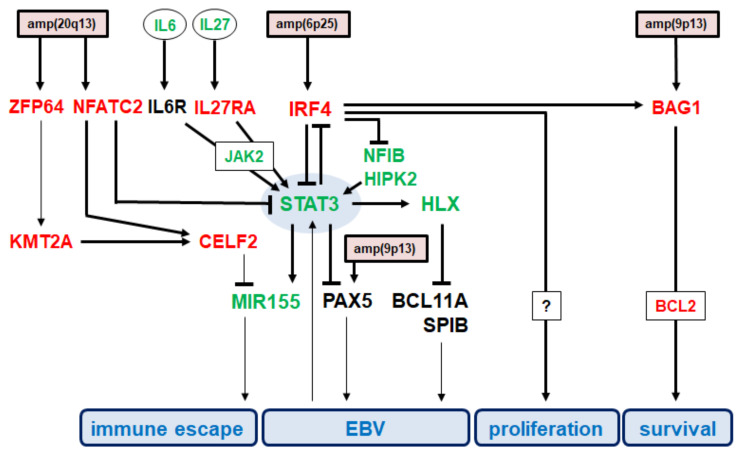
Summary of the results from this study showing a gene regulatory network around STAT3. Genomic amplifications targeting particular genes are highlighted by a red background. Elevated genes are indicated in red, low level expressed genes in green while medium level expressed genes are shown in black. Functional consequences including immune escape, EBV activity, proliferation and survival are highlighted by a blue background. Bold lines indicate relationships demonstrated in this study while slight lines refer to published data.

## Data Availability

Data are contained within the article or Appendix A.

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
