# Peer review of "Downregulation of STAT3 in Epstein-Barr Virus-Positive Hodgkin Lymphoma"

_biomedicines, 2022, doi:10.3390/biomedicines10071608_

Round 1
Reviewer 1 Report
I believe that identifying key oncogene STAT3 regulation caused by Epstein Barr Virus infection of HL is an important study. The paper is well-written, with sections that are properly delineated and adequate information for publishing. This paper will be ready for final publication after minor revisions. I feel the subject of this post to be adequate and significant. I suggest adopting this evaluation with a few small changes. As the work is generally well-written, I suggest it be accepted for publication in Frontiers in Pharmacology with a few minor revisions.
1. First of all, the majority of Figure 1.B is distinguished by dyeing, but the clumpy is not identified.
2. And Figure 2, authors must include a scale bar, it should be replaced by a clear imaging data in order for the arrow to convey the intended meaning. And each image's experimental settings must be identical, but DAPI must appear differently, which is a significant distinction. It should be replaced with different outcomes.
Author Response
Response to the Reviewers
We would like to thank the Reviewers for their helpful comments to improve the quality of our study. Here, we give detailed answers to their comments.
Reviewer 1
- We have corrected the legend of Figure 1 accordingly. To clarify the procedure we have indicated that the picture was taken by phase-contrast mikroscopy. The description of cell clumps was changed into cell aggregates.
- We have corrected Figure 2 including its legend accordingly. The images were provided with scale bars. To clarify the different treatments in Figure 2B we added the descriptions “untreated“ and “transfected“. Furthermore, we added an additional arrowhead, indicating a non-transfected cell.
Reviewer 2
We have added all names and accession numbers of the used cell lines in the Materials and Methods section 2.1.
We hope that we have answered all critical questions and comments and that our manuscript is now suitable for publication in Biomedicines.
Sincerely, Stefan Nagel
Reviewer 2 Report
Dear authors,
It was my pleasure to review your manuscript. It is a outstanding work and please find my below comments.
In the herein manuscript, entitled “Downregulation of STAT3 in Epstein Barr Virus Positive Hodgkin Lymphoma”, the authors investigated the L-540 cell line, a model of HL, in which they recently showed a constitutively activated STAT 3. The authors also used AM-HLH cells which showed a decreased expression of STAT3. The results presented in this manuscript are strong and support the hypothesis. Am-HLH cell line represents a model to study the roles and implications of STAT3 and EBV in HL.
The introductions are briefly presenting the Hodgkin Lymphoma and its subtypes. Furthermore, the molecular information summarizing relevant literature is following the subject of this study. I have no other observations.
The materials and methods are well described, easy to follow and other researchers could easily use the information for other studies. One minor point at this chapter: 2.1. Cell lines and treatment – I think that it will be useful if the cell lines names and IDs are mentioned in the text. The culture conditions and other information are ok to be just mentioned as other literature sources.
At the results section, the data is comprehensively presented and the results are sustaining the hypothesis of this study. Solid molecular data is presented and the quality of the study design is impressive. At this chapter I have no observations.
In the discussion sections, the schematic representation of the results from gene expression experiments is welcomed. Figure 6 is a good summary for the above presented results. At this section I have no observations.
The conclusion is short, and it summarizes the results. Furthermore, the conclusion is sustained by the experimental data.
The references are updated, I detected no irrelevant auto citations. Also, no plagiarism was detected.
I don't feel qualified to judge about the English language and style
I recommend the publication of this manuscript with minor changes that I suggest at the materials and methods section if the authors find it useful, if not, the paper could be accepted in present form.
Author Response

(The authors gave the same response as above.)
